# Multiview Aggregation for Learning Category-Specific Shape Reconstruction

**Srinath Sridhar**[1]  **Davis Rempe**[1]  **Julien Valentin**[2]  **Sofien Bouaziz**[2]  **Leonidas J. Guibas**[1,3]

[1]Stanford University   [2]Google Inc.   [3]Facebook AI Research

✉ ssrinath@cs.stanford.edu

🌐 geometry.stanford.edu/projects/xnocs

## Abstract

We investigate the problem of learning category-specific 3D shape reconstruction from a variable number of RGB views of previously unobserved object instances. Most approaches for multiview shape reconstruction operate on sparse shape representations, or assume a fixed number of views. We present a method that can estimate dense 3D shape, and aggregate shape across multiple and varying number of input views. Given a single input view of an object instance, we propose a representation that encodes the dense shape of the visible object surface as well as the surface behind line of sight occluded by the visible surface. When multiple input views are available, the shape representation is designed to be aggregated into a single 3D shape using an inexpensive union operation. We train a 2D CNN to learn to predict this representation from a variable number of views (1 or more). We further aggregate multiview information by using permutation equivariant layers that promote order-agnostic view information exchange at the feature level. Experiments show that our approach is able to produce dense 3D reconstructions of objects that improve in quality as more views are added.

## 1   Introduction

Learning to estimate the 3D shape of objects observed from one or more views is an important problem in 3D computer vision with applications in robotics, 3D scene understanding, and augmented reality. Humans and many animals perform well at this task, especially for known object categories, even when observed object instances have never been encountered before [27]. We are able to infer the 3D surface shape of both object parts that are directly visible, and of parts that are occluded by the visible surface. When provided with more views of the instance, our confidence about its shape increases. Endowing machines with this ability would allow us to operate and reason in new environments and enable a wide range of applications. We study this problem of learning category-specific 3D surface shape reconstruction given a variable number of RGB views (1 or more) of an object instance.

There are several challenges in developing a learning-based solution for this problem. First, we need a representation that can encode the 3D geometry of both the visible and occluded parts of an object while still being able to aggregate shape information across multiple views. Second, for a given object category, we need to learn to predict the shape of new instances from a variable number of views at test time. We address these challenges by introducing a new representation for encoding category-specific 3D surface shape, and a method for learning to predict shape from a variable number of views in an order-agnostic manner.

Representations such as voxel grids [6], point clouds [9, 17], and meshes [11, 40] have previously been used for learning 3D shape. These representations can be computationally expensive to operate on, often produce only sparse or smoothed-out reconstructions, or decouple 3D shape from 2D

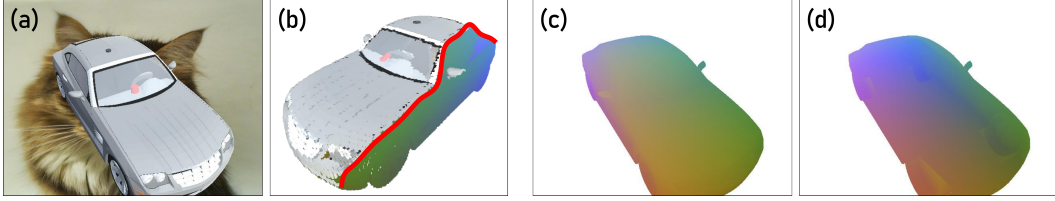

Figure 1: An input RGB view of a previously unseen object instance (a). Humans are capable of inferring the shape of the visible object surface (original colors in (b)) as well as the parts that are outside the line of sight (separated by red line in (b)). We propose an extended version of the NOCS map representation [39] to encode both the visible surface (c) and the occluded surface furthest from the current view, the **X-NOCS map** (d). Note that (c) and (d) are in exact pixel correspondence to (a), and their point set union yields the complete 3D shape of the object. RGB colors denote the XYZ position within NOCS. We learn category-specific 3D reconstruction from one or more views.

projection losing 2D–3D correspondence. To overcome these issues, we build upon the *normalized object coordinate space maps* (**NOCS maps**) representation [39]—a 2D projection of a shared category-level 3D object shape space that can encode intra-category shape variation (see Figure 1). A NOCS map can be interpreted as a 3D surface reconstruction in a canonical space of object pixels directly visible in an image. NOCS maps retain the advantages of point clouds and are implicitly grounded to the image since they provide a strong pixel–shape correspondence—a feature that allows us to copy object texture from the input image. However, a NOCS map only encodes the surface shape of object parts directly in the line of sight. We extend it to also encode the 3D shape of object parts that are occluded by the visible surface by predicting the shape of the object surface furthest and hidden from the view—called **X-NOCS maps** (see Figure 1). Given a single RGB view of an object instance, we aim to reconstruct the NOCS maps corresponding to the visible surface and the X-NOCS map of the occluded surface. Given multiple views, we aggregate the predicted NOCS and X-NOCS maps from each view into a single 3D shape using an inexpensive union operation.

To learn to predict these visible and occluded NOCS maps for one or more views, we use an encoder-decoder architecture based on SegNet [3]. We show that a network can learn to predict shape independently for each view. However, independent learning does not exploit multiview overlap information. We therefore propose to aggregate multiview information in a view order-agnostic manner by using *permutation equivariant layers* [43] that promote information exchange among the views at the feature level. Thus, our approach **aggregates multiview information** both at the *shape level*, and at the *feature level* enabling better reconstructions. Our approach is trained on a variable number of input views and can be used on a different variable number of views at test time. Extensive experiments show that our approach outperforms other state-of-the-art approaches, is able to reconstruct object shape with fine details, and accurately captures dense shape while improving reconstruction as more views are added, both during training and testing.

## 2 Related Work

Extensive work exists on recognizing and reconstructing 3D shape of objects from images. This review focuses on learning-based approaches which have dominated recent state of the art, but we briefly summarize below literature on techniques that rely purely on geometry and constraints.

**Non-Learning 3D Reconstruction Methods**: The method presented in [22] requires user input to estimate both camera intrinsics and multiple levels of reconstruction detail using primitives which allow complete 3D reconstruction. The approach of [38] also requires user input but is more data-driven and targets class-based 3D reconstruction of the objects on the Pascal VOC dataset. [4] is another notable approach for class-based 3D reconstruction a parametric 3D model and corresponding parameters per object-instance are predicted with minimal user intervention. We now focus on learning-based methods for single and multiview reconstruction.

**Single-View Reconstruction**: Single-view 3D reconstruction of objects is a severely under-constrained problem. Probabilistic or generative techniques have been used to impose constraints on the solution space. For instance, [21] uses structure from motion to estimate camera parameters

and learns category-specific generative models. The approach of [9] learns a generative model of un-ordered point clouds. The method of [15] also argue for learning generative models that can predict 3D shape, pose and lighting from a single image. Most techniques implicitly or explicitly learn class-specific generative models, but there are some, e.g., [36], that take a radically different approach and use multiple views of the same object to impose a geometric loss during training. The approach of [42] predicts 2.5D sketches in the form of depth, surface normals, and silhouette images of the object. It then infers the 3D object shape using a voxel representation. In [13], the authors present a technique that uses silhouette constraints. That loss is not well suited for non-convex objects and hence the authors propose to use another set of constraints coming from a generative model which has been taught to generate 3D models. Finally, [44] propose an approach that first predicts depth from a 2D image which is then projected onto a spherical map. This map is inpainted to fill holes and backprojected into a 3D shape.

**Multiview Reconstruction**: Multiple views of an object add more constraints to the reconstructed 3D shape. Some of the most popular constraints in computer vision are multiview photometric consistency, depth error, and silhouette constraints [18, 41]. In [20], the authors assume that the pose of the camera is given and extract image features that are un-projected in 3D and iteratively fused with the information from other views into a voxel grid. Similarly, [16] uses structure from motion to extract camera calibration and pose. [23] proposes an approach to differentiable point-cloud rendering that effectively deals with the problem of visibility. Some approaches jointly perform the tasks of estimating the camera parameters as well as reconstructing the object in 3D [17, 45].

**Permutation Invariance and Equivariance**: One of the requirements of supporting a variable number of input views is that the network must be agnostic to the order of the inputs. This is not the case with [6] since their RNN is sensitive to input view order. In this work, we use ideas of permutation invariance and equivariance from DeepSets [29, 43]. Permutation invariance has been used in computer vision in problems such as burst image deblurring [2], shape recognition [35], and 3D vision [28]. Permutation equivariance is not as widely used in vision but is common in other areas [29, 30]. Other forms of approximate equivariance have been used in multiview networks [7]. A detailed theoretical analysis is provided by [25].

**Shape Representations**: There are two dominant families of shape representations used in literature: volumetric and surface representations, each with their trade-offs in terms of memory, closeness to the actual surface and ease of use in neural networks. We offer a brief review and refer the reader to [1, 34] for a more extensive study.

The voxel representation is the most common volumetric representation because of its regular grid structure, making convolutional operators easy to implement. As illustrated in [6] which performs single and multiview reconstructions, voxels can be used as an occupancy grid, usually resulting in coarse surfaces. [26] demonstrates high quality reconstruction and geometry completion results. However, voxels have high memory cost, especially when combined with 3D convolutions. This has been noted by several authors, including [31] who propose to first predict a series of 6 depth maps observed from each face of a cube containing the object to reconstruct. Each series of 6 depth map represent a different surface, allowing to efficiently capture both the outside and the inside (occluded) parts of objects. These series of depth maps are coined shape layers and are combined in an occupancy grid to obtain the final reconstructions.

Surface representations have advantages such as compactness, and are amenable to differentiable operators that can be applied on them. They are gaining popularity in learning 3D reconstruction with works like [19], where the authors present a technique for predicting category-specific mesh (and texture) reconstructions from single images, or explorations like in [9], which introduces a technique for reconstructing the surface of objects using point clouds. Another interesting representation is scene coordinates which associates each pixel in the image with a 3D position on the surface of the object or scene being observed. This representation has been successfully used for several problems including camera pose estimation [37] and face reconstruction [10]. However, it requires a scene- or instance-specific scan to be available. Finally, geometry images [12] have been proposed to encode 3D shape in images. However, they lack input RGB pixel to shape correspondence.

In this work, we propose a *category-level surface representation* that has the advantages of point clouds but encodes strong pixel–3D shape correspondence which allows multiview shape aggregation without explicit correspondences.

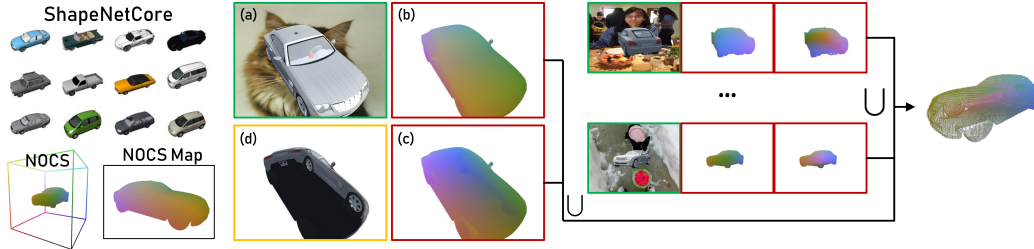

Figure 2: Given canonically aligned and scaled instances from an object category [5], the NOCS representation [39] can be used to encode intra-category shape variation. For a single view (a), a NOCS map encodes the shape of the visible parts of the object (b). We extend this representation to also encode the occluded parts called an X-NOCS map (c). Multiple (X-)NOCS maps can be trivially combined using a set union operation (⋃) into a single dense shape (rightmost). We can also efficiently represent the texture of object surfaces that are not directly observable (d). Inputs to our method are shown in green boxes, predictions are in red, and optional predictions are in orange.

## 3 Background and Overview

In this section, we provide a description of our shape representation, relevant background, and a general overview of our method.

**Shape Representation**: Our goal is to design a shape representation that can capture dense shapes of both the visible and occluded surfaces of objects observed from any given viewpoint. We would like a representation that can support computationally efficient signal processing (e.g., 2D convolution) while also having the advantages of 3D point clouds. This requires a strong coupling between image pixels and 3D shapes. We build upon the **NOCS map** [39] representation, which we describe below.

The Normalized Object Coordinates Space (NOCS) can be described as the 3D space contained within a unit cube as shown in Figure 2. Given a collection of shapes from a category which are consistently oriented and scaled, we build a shape space where the *XYZ* coordinates within NOCS represent the shape of an instance. A **NOCS map** is a 2D projection of the 3D NOCS points of an instance as seen from a particular viewpoint. Each pixel in the NOCS map denotes the 3D position of that object point in NOCS (color coded in Figure 2). NOCS maps are dense shape representations that scale with the size of the object in the view—objects that are closer to the camera with more image pixels are denser than object further away. They can readily be converted to a point cloud by reading out the pixel values, but still retain 3D shape–pixel correspondence. Because of this correspondence we can obtain camera pose in the canonical NOCS space using the direct linear transform algorithm [14]. However, NOCS maps only encode the shape of the visible surface of the object.

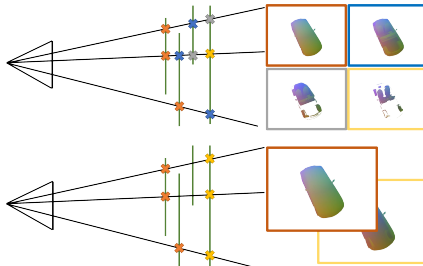

Figure 3: We use depth peeling to extract X-NOCS maps corresponding to different ray intersections. The top row shows 4 intersections. The bottom row shows our representation which uses the first and last intersections.

**Depth Peeling**: To overcome this limitation and encode the shape of the occluded object surface, we build upon the idea of depth peeling [8] and layered depth images [33]. Depth peeling is a technique used to generate more accurate order-independent transparency effects when blending transparent objects. As shown in Figure 3, this process refers to the extraction of object depth or, alternatively, NOCS coordinates corresponding to the $k^{th}$ intersection of a ray passing through a given image pixel. By peeling a sufficiently large number of layers (e.g., $k = 10$), we can accurately encode the interior and exterior shape of an object. However, using many layers can be unnecessarily expensive, especially if the goal is to estimate only the external object surface. We therefore propose to use 2 layers to approximate the external surfaces corresponding the first and last ray intersections. These intersections faithfully capture the visible and occluded parts of most common convex objects. We

refer to the maps corresponding to the occluded surface (i.e., last ray intersection) as **X-NOCS maps**, similar to X-ray images.

Both NOCS and X-NOCS maps support multiview shape aggregation into a single 3D shape using an inexpensive point set union operation. This is because NOCS is a canonical and normalized space where multiple views correspond to the same 3D space. Since these maps preserve pixel–shape correspondence, they also support estimation of object or camera pose in the canonical NOCS space [39]. We can use the direct linear transform [14] to estimate camera pose, up to an unknown scale factor (see supplementary document). Furthermore, we can support the prediction of the texture of the occluded parts of the object by hallucinating a peeled color image (see Figure 2 (d)).

**Learning Shape Reconstruction**: Given the X-NOCS map representation that encodes the 3D shape both of occluded object surfaces, our goal is to learn to predict both maps from a variable number of input views and aggregate multiview predictions. We adopt a supervised approach for this problem. We generated a large corpus of training data with synthetic objects from 3 popular categories—cars, chairs, and airplanes. For each object we render multiple viewpoints, as well the corresponding ground truth X-NOCS maps. Our network learns to predict the (X-)NOCS maps corresponding to each view using a SegNet-based [3] encoder-decoder architecture. Learning independently on each view does not exploit the available multiview overlap information. We therefore aggregate multiview information at the feature level by using permutation equivariant layers that combine input view information in an order-agnostic manner. The multiview aggregation that we perform at the NOCS shape and feature levels allows us to reconstruct dense shape with details as we show in Section 5.

# 4 Method

Our goal is to learn to predict the both NOCS and X-NOCS maps corresponding to a variable number of input RGB views of previously unobserved object instances. We adopt a supervised learning approach and restrict ourselves to specific object categories. We first describe our general approach to this problem and then discuss how we aggregate multiview information.

## 4.1 Single-View (X-)NOCS Map Prediction

The goal of this task is to predict the NOCS maps for the visible ($N_v$) and X-NOCS maps for the occluded parts ($N_o$) of the object given a single RGB view $I$. We assume that no other multiview inputs are available at train or test time. For this pixel-level prediction task we use an encoder-decoder architecture similar to SegNet [3] (see Figure 4). Our architecture takes a 3 channel RGB image as input and predicts 6 output channels corresponding to the NOCS and X-NOCS maps ($N^i = \{N_v, N_o\}$), and optionally also predicts a peeled color map ($C_p$) encoding the texture of the occluded object surface (see Figure 2 (d)). We include skip connections between the encoder and decoder to promote information sharing and consistency. To obtain the 3D shape of object instances, the output (X-)NOCS maps are combined into a single 3D point cloud as $P = \mathcal{R}(N_v) \bigcup \mathcal{R}(N_o)$, where $\mathcal{R}$ denotes a readout operation that converts each map to a 3D point set.

**(X-)NOCS Map Aggregation**: While single-view (X-)NOCS map prediction is trained independently on each view, it can still be used for multiview shape aggregation. Given multiple input views, $\{I_0, \ldots, I_n\}$, we predict the (X-)NOCS maps $\{N^0, \ldots, N^n\}$ for each view *independently*. NOCS represents a canonical and normalized space and thus (X-)NOCS maps can also be interpreted as dense correspondences between pixels and 3D NOCS space. Therefore any set of (X-)NOCS maps will map into the same space—multiview consistency is implicit in the representation. Given multiple independent (X-)NOCS maps, we can combine them into a single 3D point cloud as $P_n = \bigcup_{i=0}^{n} \mathcal{R}(N^i)$.

**Loss Functions**: We experimented with several loss functions for (X-)NOCS map prediction including a pixel-level $L^2$ loss, and a combined pixel-level mask and $L^2$ loss. The $L^2$ loss is defined as

$$\mathcal{L}_e(\mathbf{y}, \hat{\mathbf{y}}) = \frac{1}{n} \sum ||\mathbf{y} - \hat{\mathbf{y}}||_2, \ \forall \mathbf{y} \in N_v, N_o, \forall \hat{\mathbf{y}} \in \hat{N}_v, \hat{N}_o, \tag{1}$$

where $\mathbf{y}, \hat{\mathbf{y}} \in \mathbb{R}^3$ denote the ground truth and predicted 3D NOCS value, $\hat{N}_v, \hat{N}_o$ are the predicted NOCS and X-NOCS maps, and $n$ is the total number of pixels in the X-NOCS maps. However, this function computes the loss for all pixels, even those that do not belong to the object thus wasting

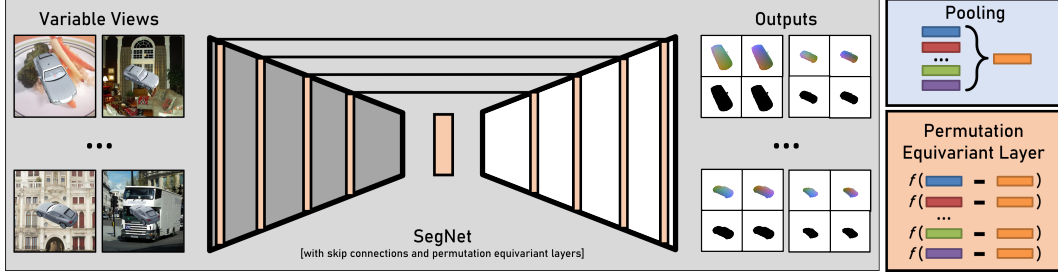

Figure 4: We use an encoder-decoder architecture based on SegNet [3] to predict NOCS and X-NOCS maps from an input RGB view independently. To better exploit multiview information, we propose to use the same architecture but with added permutation equivariant layers (bottom right) to combine multiview information at the feature level. Our network can operate on a variable number of input views in an order-agnostic manner. The features extracted for each view during upsampling and downsampling operations are combined using permutation equivariant layers (orange bars).

network capacity. We therefore use object masks to restrict the loss computation only to the object pixels in the image. We predict 2 masks corresponding to the NOCS and X-NOCS maps—8 channels in total. We predict 2 independent masks since they could be different for thin structures like airplane tail fins. The combined mask loss is defined as $\mathcal{L}_m = \mathcal{L}_v + \mathcal{L}_o$, where the loss for the visible NOCS map and mask is defined as

$$\mathcal{L}_v(\mathbf{y}, \hat{\mathbf{y}}) = w_m \, \mathcal{M}(M_v, \hat{M}_v) + w_l \, \frac{1}{m} \sum ||\mathbf{y} - \hat{\mathbf{y}}||_2, \ \forall \mathbf{y} \in M_v, \forall \hat{\mathbf{y}} \in \hat{M}_v, \qquad (2)$$

where $\hat{M}_v$ is the predicted mask corresponding to the visible NOCS map, $M_v$ is the ground truth mask, $\mathcal{M}$ is the binary cross entropy loss on the mask, and $m$ is the number of masked pixels. $\mathcal{L}_o$ is identical to $\mathcal{L}_v$ but for the X-NOCS map. We empirically set the weights $w_m$ and $w_l$ to be 0.7 and 0.3 respectively. Experimentally, we observe that the combined pixel-level mask and $L^2$ loss outperforms the $L^2$ loss since more network capacity can be utilized for shape prediction.

## 4.2 Multiview (X-)NOCS Map Prediction

The above approach predicts (X-)NOCS maps independently and aggregates them to produce a 3D shape. However, multiview images of an object have strong inter-view overlap information which we have not made use of. To promote information exchange between views both during training and testing, and to support a variable number of input views, we propose to use permutation equivariant layers [43] that are agnostic to the order of the views.

**Feature Level Multiview Aggregation**: Our multiview aggregation network is illustrated in Figure 4. The network is identical to the single-view network except for the addition of several permutation equivariant layers (orange bars). A network layer is said to be permutation equivariant if and only if the off diagonal elements of the learned weight matrix are equal, as are the diagonal elements [43]. In practice, this can be achieved by passing each feature map through a *pool-subtract* operation followed by a non-linear function. The pool-subtract operation pools features extracted from different viewpoints and subtracts the pooled feature from the individual features (see Figure 4). We use multiple permutation equivariant layers after each downsampling and upsampling operation in the encoder-decoder architecture (vertical orange bars in Figure 4). Both average pooling and max pooling can used but experimentally average pooling worked best. Our permutation equivariant layers consist of an average-subtraction operation and the non-linearity from the next convolutional layer.

**Hallucinating Occluded Object Texture**: As an additional feature, we train both our single and multiview networks to also predict the texture of the occluded surface of the object (see Figure 2 (d)). This is predicted as 3 additional output channels with the same loss as $\mathcal{L}_v$. This optional prediction can be used to hallucinate the texture of hidden object surfaces.

# 5 Experiments

**Dataset**: We generated our own dataset, called *ShapeNetCOCO*, consisting of object instances from 3 categories commonly used in related work: chairs, cars, and airplanes. We use thousands of instances from the ShapeNet [5] repository and render 20 different views for each instance and additionally augment backgrounds with randomly chosen COCO images [24]. This dataset is harder than previously proposed datasets because of random backgrounds, and widely varying camera distances. To facilitate comparisons with previous work [6, 17], we also generated a simpler dataset, called *ShapeNetPlain*, with white backgrounds and 5 views per object following the camera placement procedure of [17]. Except for comparisons and Table 3, we report results from the more complex dataset. We follow the train/test protocol of [36]. Unless otherwise specified, we use a batch size of 1 (multiview) or 2 (single-view), a learning rate of 0.0001, and the Adam optimizer.

**Metrics**: For all experiments, we evaluate point cloud reconstruction using the 2-way Chamfer distance multiplied by 100. Given two point sets $S_1$ and $S_2$ the Chamfer distance is defined as

$$d(S_1, S_2) = \frac{1}{|S_1|} \sum_{x \in S_1} \min_{y \in S_2} \|x - y\|_2^2 + \frac{1}{|S_2|} \sum_{y \in S_2} \min_{x \in S_1} \|x - y\|_2^2. \tag{3}$$

## 5.1 Design Choices

Table 1: Single-view reconstruction performance using various losses and outputs. For each category, the Chamfer distance is shown. Using the joint loss with $L^2$ and the mask significantly outperforms just $L^2$. Predicting peeled color further improves reconstruction.

| Loss | Output | Cars | Airplanes | Chairs |
|------|--------|------|-----------|--------|
| L2 | (X-)NOCS+Peel | 3.6573 | 7.9072 | 4.4716 |
| L2+Mask | (X-)NOCS+Mask | 0.5093 | 0.3037 | 0.4401 |
| L2+Mask | (X-)NOCSS+Mask+Peel | **0.3714** | **0.2659** | **0.4288** |

We first justify our loss function choice and network outputs. As described, we experiment with two loss functions—$L^2$ losses with and without a mask. Further, there are several outputs that we predict in addition to the NOCS and X-NOCS maps i.e., mask and peeled color. In Table 1, we summarize the average Chamfer distance loss for all variants trained independently on single views (*ShapeNetCOCO* dataset). Using the loss function which jointly accounts for NOCS map, X-NOCS maps and mask output clearly outperforms a vanilla $L^2$ loss on the NOCS and X-NOCS maps. We also observe that predicting peeled color along with the (X-)NOCS maps gives better performance on all categories.

## 5.2 Multiview Aggregation

Next we show that our multiview aggregation approach is capable of estimating better reconstructions when more views are available (*ShapeNetCOCO* dataset). Table 2 shows that the reconstruction from the single view network improves as we aggregate more views into NOCS space (using set union) without any feature space aggregation. When we train with feature space aggregation from 5 views using the permutation equivariant layers we see further improvements as more views are added. Table 3 shows

Table 2: Comparison of different forms of multiview aggregation. Aggregating multiple views using set union improves performance with further improvements using feature space aggregation.

| Category | Model | 2 views | 3 views | 5 views |
|----------|-------|---------|---------|---------|
| Cars | Single-View | 0.4206 | 0.3974 | 0.3692 |
| | Multiview | **0.3789** | **0.3537** | **0.2731** |
| Airplanes | Single-View | **0.1760** | **0.1677** | 0.1619 |
| | Multiview | 0.2387 | 0.1782 | **0.1277** |
| Chairs | Single-View | 0.4249 | 0.3813 | 0.3600 |
| | Multiview | **0.3649** | **0.2860** | **0.2457** |

variations of our multiview model: one trained on a fixed number of views, one trained on a variable number of views up to a maximum of 5, and one trained on a variable number up to 10 views. All these models are trained on the *ShapeNetPlain* dataset for 100 epochs. We see that both fixed and variable models take advantage of the additional information from more views, almost always increasing performance from left to right. Although the fixed multiview models perform best, we hypothesize that the variable view models will be able to better handle the widening gap between the number of train-time and test-time views. In Figure 5, we visualize our results in 3D which shows the small scale details such as airplane engines reconstructed by our method.

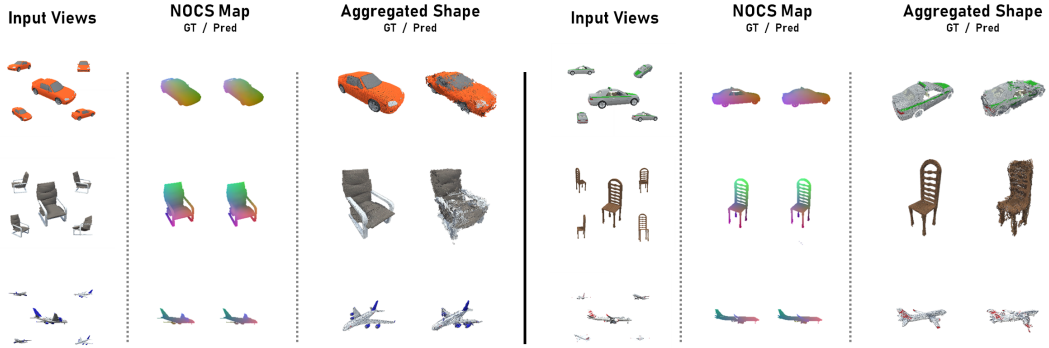

Figure 5: Qualitative reconstructions produced by our method. Each rows shows the input RGB views, NOCS map ground truth and prediction of the central view, and the ground truth and predicted 3D shape. These visualizations are produced by the variable multiview model trained on up to 5 views and evaluated on 5 views. We post-process the both NOCS and X-NOCS maps with a bilateral filter followed by a statistical outlier filter [32], and use the input RGB images to color the point cloud. Best viewed zoomed and in color.

## 5.3 Comparisons

Table 3: Multiview reconstruction variations. We observe that both fixed and variable models take advantage of the additional information from more views.

| Category | Model | 2 views | 3 views | 5 views |
|---|---|---|---|---|
| Cars | Fixed Multi | **0.2645** | **0.1645** | **0.1721** |
| | Variable Multi (5) | 0.2896 | 0.1989 | 0.1955 |
| | Variable Multi (10) | 0.2992 | 0.2447 | 0.3095 |
| Airplanes | Fixed Multi | **0.1318** | 0.1571 | **0.0604** |
| | Variable Multi (5) | 0.1418 | **0.1006** | 0.0991 |
| | Variable Multi (10) | 0.1847 | 0.1309 | 0.1049 |
| Chairs | Fixed Multi | 0.2967 | **0.1845** | **0.1314** |
| | Variable Multi (5) | **0.2642** | 0.2072 | 0.1695 |
| | Variable Multi (10) | 0.2643 | 0.2070 | 0.1693 |

We compare our method to two previous works. The first, called differentiable point clouds (DPC) [17] directly predicts a point cloud given a single image of an object. We train a separate single-view model for cars, airplanes, and chairs to predict the NOCS maps, X-NOCS maps, mask and peeled color (*ShapeNetPlain* dataset). To evaluate the Chamfer distance for DPC outputs, we first scale the predicted output point cloud such that the bounding box diagonal is one, then we follow the alignment procedure from their paper to calculate the transformation from the network's output frame to the ground truth point cloud frame. As seen in Table 4, the X-NOCS map representation allows our network to outperform DPC in all three categories.

We next compare our multiview permutation equivariant model to the multiview method 3D-R2N2 [6]. In each training batch, both methods are given a random subset of 5 views of an object, so that they may be evaluated with up to 5 views at test time. Since 3D-R2N2 outputs a volumetric 32x32x32 voxel grid, we first find all surface voxels of the output then place a point at the center of these surface voxels to obtain a 3D point cloud. This point cloud is scaled to have a unit-diagonal bounding box to match the ground truth ShapeNet objects. We limit our comparison to only chairs since we were unable to make their method converge on the other categories.

Table 4: Single-view reconstruction comparison to DPC [17].

| Method | Cars | Airplanes | Chairs |
|---|---|---|---|
| DPC | 0.2932 | 0.2549 | 0.4314 |
| Ours | **0.1569** | **0.1855** | **0.3803** |

Table 5: Multiview reconstruction performance compared to 3D-R2N2 [6] on the chairs category in *ShapeNetPlain*.

| Method | 2 views | 3 views | 5 views |
|---|---|---|---|
| 3D-R2N2 | 0.2511 | 0.2191 | 0.1932 |
| Ours | **0.2508** | **0.1952** | **0.1576** |

Table 5 shows the performance of both methods when trained on the chairs category and evaluated on 2, 3, and 5 views (*ShapeNetPlain* dataset). For 2 views the methods perform similar but when combining more views to reconstruct the shape, our method becomes more accurate. We again see the trend of increasing performance as more views are used.

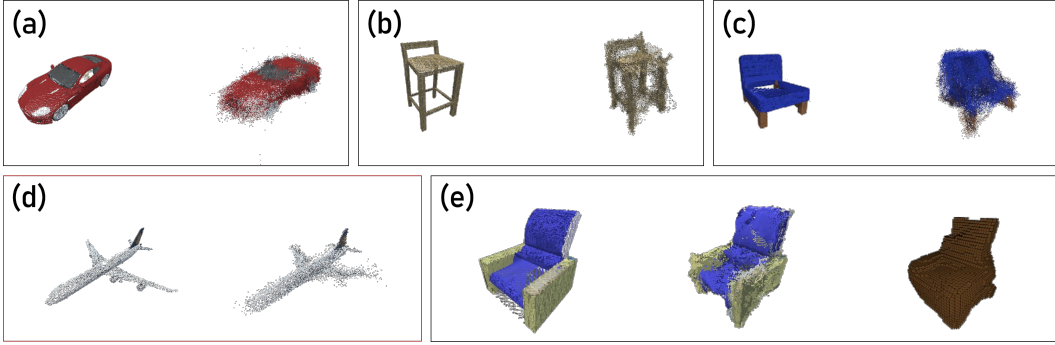

Figure 6: More qualitative reconstructions produced by our method. For each box, ground truth is shown leftmost. Here we show reconstructions from (a) the permutation equivariant network trained and tested on 10 views for a car, and (b, c) permutation equivariant network trained on chairs with 5 views and tested on 5. A reconstruction with higher shape variance that fails to capture small scale detail is shown in (d). Finally, in (e) we show a visual comparison with the reconstruction produced by [6] which lacks detail such as the armrest although it sees 5 different views. Best viewed in color.

**Limitations and Future Work**: While we reconstruct dense 3D shapes, there is still some variance in our predicted shape. We can further improve the quality of our reconstructions by incorporating surface topology information. We currently use the DLT algorithm [14] to predict camera pose in our canonical NOCS space, however we would need extra information such as depth [39] to estimate metric pose. Jointly estimating pose and shape is a tightly coupled problem and an interesting future direction. Finally, we observed that Chamfer distance, although used widely to evaluate shape reconstruction quality, is not the ideal metric to help differentiate fine scale detail and overall shape. We plan to explore the use of the other metrics to evaluate reconstruction quality.

## 6   Conclusion

In this paper we introduced X-NOCS maps, a new and efficient surface representation that is well suited for the task of 3D reconstruction of objects, even of occluded parts, from a variable number of views. We demonstrate how this representation can be used to estimate the first and the last surface point that would project on any pixel in the observed image, and also to estimate the appearance of these surface points. We then show how adding a permutation equivariant layer allows the proposed method to be agnostic to the number of views and their associated viewpoints, but also how our aggregation network is able to efficiently combine these observations to yield even higher quality results compared to those obtained with a single observation. Finally, extensive analysis and experiments validate that our method reaches state-of-the-art results using a single observation, and significantly improves upon existing techniques.

**Acknowledgments**: This work was supported by the Google Daydream University Research Program, AWS Machine Learning Awards Program, and the Toyota-Stanford Center for AI Research. We would like to thank Jiahui Lei, the anonymous reviewers, and members of the Guibas Group for useful feedback. Toyota Research Institute ("TRI") provided funds to assist the authors with their research but this article solely reflects the opinions and conclusions of its authors and not TRI or any other Toyota entity.

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
