[Supplementary Material]

# Multiview Aggregation for Learning Category-Specific Shape Reconstruction

## 1 Supplementary Document

In this supplementary document, we show more qualitative results of our reconstructions, and raw NOX maps predicted by our network. We additionally also show that our approach can support articulating object categories like the hand, and show example camera pose estimation results. We will make all datasets and code public upon publication of the paper.

**Dataset**: Figure 1 shows some sample data from our synthetic dataset from each of the 3 categories we consider. As opposed to previous datasets, our dataset features challenging backgrounds, and widely varying camera pose and distance to objects.

Figure 1: Two sample views of an instance from each category in our dataset. Our dataset features widely camera viewpoints and backgrounds which are more challenging than previous datasets.

**NOX Maps**: Figure 2 shows a zoomed-in version of the ground truth and predicted NOX maps (i.e., visible and occluded NOCS maps). Our predictions are able to capture the dense 3D shape of the visible and occluded object surfaces while maintaining pixel-shape correspondence.

**Qualitative Results**: In Figure 3, we show more detailed qualitative results including the predicted NOX maps and peeled color maps. Please see the main paper for details on each of the results.

**Articulating Category**: Our approach is generalizable to other object categories. In particular, we show that we can extend our approach to an articulating category like the human hand. We generated a dataset with a virtual hand animated to various poses and augmented with background images. We generated 5 views for each pose. We train the single-view NOCS map prediction network (only the

Figure 2: For the input RGB view on the left, we predict a NOX map i.e., NOCS maps for the visible and occluded object surfaces. For each box on the right, the grounth truth is shown in the left and our prediction from the permutation equivariant network on the right.

Figure 3: Detailed qualitative results for the reconstructions shown in Figure 5 in the main paper. Here, for each example, we show two views with the corresponding NOX maps (ground truth on left, prediction on right), peeled color map, and the 3D point cloud (ground truth on top, reconstruction on bottom).

18   visible NOCS map) with the combined mask and $L^2$ loss function. Figure 4 shows a test pose from
19   the validation set with previously unseen hand shape and pose. Figure 5 shows the 3D point cloud
20   reconstructions of the predicted NOCS maps.

21   **Camera Pose Estimation**: Finally, we demonstrate that our approach can also be used for estimating
22   camera pose from the predicted NOCS maps. We use the direct linear transform algorithm [1] to
23   obtain camera intrinsics and extrinsics parameters in the NOCS space. Thus, the estimated pose is
24   accurate upto an unknown scale factor. We leave evaluation of camera pose for future work.

Figure 4: Reconstruction of an articulating shape category—hands. Here we trained a single-view network to predict only the visible NOCS map.

Figure 5: Example of camera pose estimation from the predicted NOCS maps for the hands dataset. For each view, we use th DLT algorithm to obtain camera pose in the canonical NOCS space (axes colored red-green-blue). We can then use the set union operation to aggregate the 3D shape.

# References

[1] Richard Hartley and Andrew Zisserman. *Multiple view geometry in computer vision*. Cambridge university press, 2003.