[Reviews · NeurIPS 2019]

Reviewer 1



Summary: The authors address the problem of learning category-specific shape reconstruction using the proposed NOX representation. The NOX representation builds on the NOCS idea of normalized object coordinate systems which represents all instances in an object category within a unit cube. Predicting a perspective projection of the NOCS representation in the camera view (called the NOCS map) is thus equivalent to predicting the object shape coordinates in the unit cube (or NOCS). The authors extend this to not just predict NOCS coordinates of the visible surface in a camera view (first intersection of ray from pixel to object) but also coordinates of the the "last* intersection of the ray. This pair of first and last intersection maps termed NOX thus provide a reasonably complete picture of object shape (for mostly convex objects). Moreover, predictions in the NOCS space can be trivially accumulated using a set union operation. The authors present experiments on predicting object shape via NOX on synthetic data generated from 3 categories in ShapeNet (cars, chairs and aeroplanes) from single and multiple views using permutation equivariant layers for the latter and compare against prior 3D shape prediction methods which predict point clouds and voxels. There are certain details missing from the experiments and certain inconsistent performance trends (elaborated on below) which require explanations/clarifications. There is also a lack of visual comparisons. Originality: The NOX representation presented in the paper is definitely novel and useful for predicting object shape beyond whats visible in the image. The idea of permutation equivariance for predicting sets of NOX maps is also novel. Quality: The core concepts presented in the paper are solid - a normalized coordinate system for representing all instances of an object category and deriving a predictable representation from it which carries benefits of 3D-2D correspondence. However, the experiments and method have certain missing details which need elaboration. - Is there a separate network for every object category or are the networks shared? If separate, do the authors believe this is because predicting NOCS maps requires specialization to object category. My issue is at this point, it boils down to a fine grained recognition task rather than a reconstruction task as detailed in [A]. Can the authors comment on this? - In L246, the authors mention comparing against a ground truth point set. How is this point set generated from Shapenet meshes? - Is the multi-view NOX prediction network (Sec 4.2) finetuned from a pre-trained single view network? - Is there any threshold for removing duplicates during the union operation? Are all the points from various views retained for multi-view reconstructions or are they filtered in any way? - In Table 2, the reconstruction quality worsens for cars in the multi-view setting with 5 views. Can the authors comment on why this happens? - Table 3 contains a row with "Fixed Multi" views. How many views were the models trained with in this setting? - In table 3, different categories have different model variants that are the best performing? The variable multi (5) with 2 views is much worse than either fixed multi and variable multi (10) for airplanes while for cars variable multi(5) is the best of the lot. Can the authors explain the reason for this variation? - The errors for chairs in Table 4 and Table 5 seem inconsistent. The authors report single view reconstruction number of 0.4637 for chairs in Table 4 and 1.0896 in Table 5. What is the reason for this difference? Are the datasets different? If so how? - The 3D-R2N2 baseline where the predicted voxels are converted into a point cloud by just sampling the centers of the grid locations seems unfair. Why not find the 0.5 probability isosurface and sample points on it for a baseline? Clarity: The paper is written well in general. Core concepts are explained clearly and diagrams are informative. However, there is a lack of visual results and comparisons. Figure 5 is the only one that shows visual results each for a different set of networks along with one in the supplementary material. However none of them show results from the method the authors compare against. There are certain lines of work that the authors miss in related work. - There is a close relation of the NOCS representation and position maps [B] / geometry images [C] used to represent surfaces in normalized spaces. Can the authors please add a discussion in related work. - Layered depth images are closely related to the back surface prediction and should be cited. - In L34, [31] is not the correct reference for methods predicting meshes as [31] still operates in voxel space. More appropriate references for this would be AtlasNet[E], Pixel2Mesh[F] etc. - Minor: reference [6] is missing the conference/venue for publication. Significance: In my opinion, the NOCS and NOX representation provide a way of reasoning about 3D shapes while still dealing with representations defined on a 2D plane. Being able to predict such a representation from a single image/images is of great significance as it opens the door for reasoning in a shared space and enforcing geometric priors and hopefully explicit multi-view reasoning in an efficient fashion. A. What Do Single-view 3D Reconstruction Networks Learn?, Tatarchenko et al., 2019 B. Joint 3D Face Reconstruction and Dense Alignment with Position Map Regression Network, Feng at al., ECCV 2018 C. Deep Learning 3D Shape Surfaces using Geometry Images, Sinha et al, ECCV 2016 D. Layered Depth Images, Shade et al, SIGGRAPH 98 E. AtlasNet: A Papier-Mâché Approach to Learning 3D Surface Generation, Groueix et al., CVPR 2018 F. Pixel2Mesh: Generating 3D Mesh Models from Single RGB Images, Wang et al. Post rebuttal update: After having read the authors rebuttal and other reivews, I still believe the paper provides value by proposing a representation (extension of NOCS) which enables 3d shape prediction (beyond whats visible) while being able to use multiple frames at train/test time and maintain the desirable property of 3D-2D correspondences. I urge the authors to significantly improve the presentation of their experiments section and add some of the justifications mentioned in the rebuttal. I am also not completely convinced by this line in the rebuttal on discussing how this method is different from other works like position maps, geometry images etc. - "In short, the differences are direct 2D–3D correspondences, the ability to use existing CNN machinery, and implicit encoding of camera pose.". This is not completely accurate as position maps (refer to PRNet) allow all of the above! The authors are strongly encouraged to give more thought to the comparisons. Finally, I stand by my original rating of "7: a good submission" after the rebuttal.

Reviewer 2



Originality: Borderline. The components are not new, and the core NOX map is a rather straight forward extension of NOCS map. However, it is not trivial to assemble a pipeline that consistently outperforms others. So I would consider the originality of this paper on the borderline. Quality: Borderline. The performance under Chamfer distance metric is mostly better than other methods, however, there are some cases that deserve more explanations, e.g., Table 2-Car rows, Table 3-Car-first element of the third row. It seems that claims from Line 58-59 are not backed by results. Clarity: Good. Significance: Good.

Reviewer 3



I think both the NOX maps and the feature pooling over multiple input views make sense on an intuitive level. On the other hand, there is a large amount of literature in this area, and the proposed ideas are minor variations on well known concepts. The paper is also missing some important related work, and it would be good to consult a recent survey (https://arxiv.org/pdf/1906.06543.pdf) to make sure the most relevant work is discussed. One notable omission in the related work is "Matryoshka Networks: Predicting 3D Geometry via Nested Shape Layers" by Richter and Roth. Their geometry representation is related to depth peeling, but they use several nested layers which allows them to represent internal surfaces, as opposed to just front and back as in NOX. In addition, their results seem much higher quality than in the proposed approach. An other weakness of the paper is the limited comparison to only two other techniques, one quite dated (3D-R2N2). The second compared method (differentiable point clouds) does not use 3D supervision, hence this is not really a fair comparison. While I think there are some interesting details in the paper, overall I feel the paper is too limited for publication at NIPS. The results do not seem to be state of the art. The limited comparison does not allow a full assessment because the reference methods are dated or address a different problem statement (3D reconstruction without 3D supervision). Visually other techniques (such as Matryoshka networks by Richter and Roth) seem to lead to much higher quality.

[Author Response · NeurIPS 2019]

# Multiview Aggregation for Learning Category-Specific Shape Reconstruction

We thank the reviewers for their valuable comments, and are happy to see feedback such as "concepts presented in the paper are solid" (**R1**), "well demonstrated ... proposed approach outperforms other 3D shape reconstruction methods" (**R2**), and "makes sense on an intuitive level" (**R3**). The reviewers agree that NOX maps are "definitely novel ... predicting object shape beyond whats visible in the image" (**R1**), "Being able to predict such a representation from a single image/images is of great significance" (**R2**). In addition to the positives identified by the reviewers, NOX maps provide strong 2D–3D correspondences, can support articulating object categories, encode camera pose implicitly (see supplementary), and allow multiview aggregation both at the NOCS and feature-level (using permutation equivariant layers). We answer questions, address factual errors, and present more details to improve our manuscript.

**Experiments (R1, R2, R3):** We will improve the clarity of the experiments section in the final version and add additional details from below.

First, we address **R1**'s questions and concerns. In Table 2, why does the quality worsen for multiview setting for cars compared to single view? We believe that this is the result of cars having a simpler convex shape that does not benefit from feature-level multiview aggregation. We observe significant improvements for more complex shapes like chairs and airplanes for the same setting. While probabilistically sampling an isosurface in 3D-R2N2 makes sense, we chose to sample the center of each voxel for easier comparison following prior work which does the same [13]. In Table 3, different categories having different variants that perform well is likely due to specific shape distribution for each category. Similar to Table 2, we observe that more complex shapes like airplanes or chairs benefit from more views and information aggregation at both the feature and NOCS level. While for simpler shapes like cars, feature-space aggregation benefits less. The differences in chairs between Table 4 and 5 are due to different experimental setting. In particular, one is a single view model and the other is trained on up to 5 views but evaluated on 1 view. In Table 3, the "Fixed Multi" models were trained with 2, 3, or 5 views respectively.

**R2** questions the claims about feature-level aggregation in Tables 2 and 3. Our main claim is that adding more views improves test-time reconstruction (numbers get smaller from left to right in Table 3), not that training on more views improves performance (numbers do not always get smaller from variable 5 to 10). We will add this detail in the paper.

We disagree with **R3** that our reference methods do not allow fair comparison. The methods we compare against are the few methods that cover a subset of the problems we solve: variable multiview input at train/test time (3D-R2N2), and reconstruct complete 3D shape as a point cloud (3D-R2N2 and DPC). Our method further supports articulating categories and can encode camera pose. "Dated" methods are still valid prior art to compare against especially when they solve similar problems.

**Qualitative Results (R1, R2, R3):** Due to strict page limits, we did not include more qualitative results and comparisons but will include more like the images below to the final version.

**Related Work (R1, R2, R3):** Thanks for pointers to make our related work more exhaustive (we will add missing references and correct existing ones). Specifically, we will add a discussion about relationship and differences to geometry images, position maps, and back surface prediction. In short, the differences are direct 2D–3D correspondences, the ability to use existing CNN machinery, and implicit encoding of camera pose.

We thank **R3** for the reference to the review paper and to Matryoshka Networks which we will add. As mentioned in Figure 3, our approach is fully capable of representing the interior parts of an object far more compactly than Matryosha networks with only K layers. We can also trivially merge these layers using set union as opposed to recursive composition. We choose to only use the first and last intersections due to computational efficiency. Thus, we disagree with **R3**'s subjective characterization of our work as a "minor variation" of well-known concepts. Our independent discovery of a more compact encoding of shape suggests otherwise.

**Other Questions:** We do not employ any thresholding for the set union operation nor do we filter the estimated NOX maps. Our method is robust even without postprocessing but median and bilateral filter do improve (will add to supplementary document). In all experiments, we trained one network for each category as is normal practice but we also trained joint networks for all categories and are happy to include these results. We expect the results of "What Do Single-view 3D Reconstruction Networks Learn?" (**R1, R2**) to hold to our approach as we share similar encoder-decoder to other discussed work. However, we believe that we are learning an implicit distribution of shapes (albeit as NOCS maps) for each category and can model intra-categroy topology variation. The ground truth point set is the set union of the ground truth NOX maps which are sampled during the rendering process. Our multiview NOX nets are trained from scratch similar to the single-view nets.

[Meta-Review · NeurIPS 2019]

The paper presents a novel approach to predicting 3D shape of objects from image(s). The reviews were split, with R1 being rather positive, R3 quite negative and R2 marginally positive. Following the rebuttal and discussion period, this score discrepancy remained. I have read the paper, the reviews and the rebuttal, and despite some concerns raised by R3 (and in part agreed with by R1) I side with the (weak) majority vote, and feel that the paper has enough novelty and is likely to generate enough interest to be accepted. The proposed approach builds upon the previously proposed methodology (NOCS) but extends it in a substantial way, incorporating reasoning about the "back-facing" surface points of the object. It also proposes a novel way to aggregate (variable number of) multiple views for better reconstruction. The authors can improve the paper significantly just by incorporating into the final version various responses they included in the rebuttal.